# Combination of Cellulose Derivatives and Chitosan-Based Polymers to Investigate the Effect of Permeation Enhancers Added to In Situ Nasal Gels for the Controlled Release of Loratadine and Chlorpheniramine

**DOI:** 10.3390/polym15051206

**Published:** 2023-02-27

**Authors:** Prasanth Viswanadhan Vasantha, Sheri Peedikayil Sherafudeen, Mohamed Rahamathulla, Sam Thomarayil Mathew, Sandhya Murali, Sultan Alshehri, Faiyaz Shakeel, Prawez Alam, Ala Yahya Sirhan, Bhageerathy Anantha Narayana Iyer

**Affiliations:** 1Department of Pharmaceutics, Mount Zion College of Pharmaceutical Sciences and Research, Chayalode P.O. Ezhamkulam, Pathanamthitta Dist, Adoor 691556, India; 2Department of Pharmaceutics, Mar Discorous College of Pharmacy, Alathara, Sreekariyam, Thiruvananthapuram Dist, Thiruvananthapuram 695017, India; 3Department of Pharmaceutics, College of Pharmacy, King Khalid University, Al Faraa, P.O. Box 62223, Abha 61421, Saudi Arabia; 4Researcher and Medical Communication Expert, Bangalore 560076, India; 5Department of Pharmaceutical Sciences, College of Pharmacy, AlMaarefa University, Ad Diriyah 13713, Saudi Arabia; 6Department of Pharmaceutics, College of Pharmacy, King Saud University, Riyadh 11451, Saudi Arabia; 7Department of Pharmacognosy, College of Pharmacy, Prince Sattam Bin Abdulaziz University, Al-Kharj 11942, Saudi Arabia; 8Faculty of Pharmacy, Amman Arab University, Amman 11953, Jordan

**Keywords:** loratadine, chlorpheniramine, chitosan, permeation enhancer, steady state flux, enhancement ratio, permeability coefficient

## Abstract

The purpose of the study is to develop and assess mucoadhesive in situ nasal gel formulations of loratadine and chlorpheniramine maleate to advance the bioavailability of the drug as compared to its conventional dosage forms. The influence of various permeation enhancers, such as EDTA (0.2% *w*/*v*), sodium taurocholate (0.5% *w*/*v*), oleic acid (5% *w*/*v*), and Pluronic F 127 (10% *w*/*v*), on the nasal absorption of loratadine and chlorpheniramine from in situ nasal gels containing different polymeric combinations, such as hydroxypropyl methylcellulose, Carbopol 934, sodium carboxymethylcellulose, and chitosan, is studied. Among these permeation enhancers, sodium taurocholate, Pluronic F127 and oleic acid produced a noticeable increase in the loratadine in situ nasal gel flux compared with in situ nasal gels without permeation enhancer. However, EDTA increased the flux slightly, and in most cases, the increase was insignificant. However, in the case of chlorpheniramine maleate in situ nasal gels, the permeation enhancer oleic acid only showed a noticeable increase in flux. Sodium taurocholate and oleic acid seems to be a better and efficient enhancer, enhancing the flux > 5-fold compared with in situ nasal gels without permeation enhancer in loratadine in situ nasal gels. Pluronic F127 also showed a better permeation, increasing the effect by >2-fold in loratadine in situ nasal gels. In chlorpheniramine maleate in situ nasal gels with EDTA, sodium taurocholate and Pluronic F127 were equally effective, enhancing chlorpheniramine maleate permeation. Oleic acid has a better effect as permeation enhancer in chlorpheniramine maleate in situ nasal gels and showed a maximum permeation enhancement of >2-fold.

## 1. Introduction

Allergic rhinitis is a diverse condition characterized by mucosal infiltration and the action of mast cells, eosinophils, and plasma cells [1,2,3]. This is a common symptom, but the rates of treatment and preventative measures are comparatively low. Many formulations are used in allergic rhinitis (AR); the limitations associated with drug delivery systems (DDSs) present significant drawbacks. Some of the factors affecting DDSs include mucociliary clearance (MCC), anterior leakage and nasal drug volume capacity (<0.2 mL) [4].

Antihistamines, such as chlorpheniramine maleate and loratadine, are widely used for the symptomatic relief from allergic symptoms. Because it undergoes first-pass metabolism, it is absorbed relatively slowly from the gastrointestinal tract (GIT). The objective of current pharmaceutical research is to develop a novel DDS for existing drug compounds that will improve therapeutic action, reduce the incidence of adverse effects and thereby improve patient compliance. Drug platforms targeting the mucin layer in the pulmonary, nasal, ocular, vaginal, intestinal, sublingual, buccal, and rectal tissues are termed mucoadhesive drug delivery systems (MADSs), which exert both systemic and local effects [5].

A vast surface area, a porous endothelium membrane, a highly vascularized epithelium and the avoidance of first-pass metabolism, make the nasal passage a promising route for drug administration [6,7,8]. This facilitates easy administration, delivers an accurate dose, and prolongs the residence time (RT) of the drug in the nasal mucosa [9]. A drug’s relatively short RT in the nasal cavity directly impacts its bioavailability. A potential strategy to improve RT is to reduce rapid MCC using mucoadhesive formulations. However, mucoadhesive powders and conventional gels cannot extend the RT due to the following drawbacks: irritation of the nasal mucosa, inability to measure an accurate dose due to difficulty in administration, and the gritty appearance of the tissues [10,11].

In situ nasal gels have emerged as a viable alternative to mucoadhesive powders and conventional gels. The in situ gelling system ensures patient comfort, which is the primary requirement for controlled drug distribution. Other advantages of in situ gels include extended or sustained medication release [12]. Over the last few decades, innovative research has been conducted on the production of gels using good biodegradable, biocompatible, and water-soluble polymers through changes in pH, temperature, and ion concentration (Appendix A) [13]. Based on the review literature, a number of researchers have successfully prepared the in situ nasal gels by using different polymers, such as hydroxypropyl methylcellulose, carbopol, Poloxamer 407, Poloxamer 188, hydroxypropyl cellulose, polyethylene glycol, polyvinyl acetate, and methyl cellulose [9,14]. The interaction of mammalian cells with external molecules is controlled by the glycocalyx [15]. Samyn et al. produced crystalline nanofibers of polysaccharide cellulose, chitin, and chitosan with preserved native structural packing. The preparation of chitosan nanofibril networks was recently reported by means of a chitosan mild hydrolysis in the solid state [16]. Halimi C et al., by using a stress-controlled rheometer AR-2000, compared a new generation of dermal fillers for wrinkles based on chitosan to existing hyaluronic acid-based dermal fillers. An in vivo assessment for the preclinical proof of chitosan as a dermal filler was carried out in a pig model and it was found that there is dermal restoration around and in place of the subcutaneous implant [17]. Doench I et al. dissolved and evaluated a chitosan formulation with injectable cellulose nanofibers for disc nucleo-supplementation. The in situ gelation of formulations led to localization of the implant at the injection site, restoration of the viscoelastic properties of the discs, and restoration or increase in the disc height, which is of therapeutic interest to suppress or decrease back pain by preventing nerve root compression [18].

This work emphasises the importance and benefits of bioadhesive or mucoadhesive nasal drug delivery devices [14]. However, mucous membranes are natural barriers, and few agents can readily penetrate them in sufficient amounts to be effective. Therefore, considerable research has been performed in recent years in the field of penetration enhancement [19,20]. Penetration enhancers increase the ability of membranes to absorb drugs [21,22]. In this study, our main objective is to investigate the effect of permeation enhancers on the release of chlorpheniramine maleate and loratadine from chlorpheniramine maleate and loratadine in situ nasal gels.

## 2. Materials and Methods

### 2.1. Materials

Chlorpheniramine maleate (CPM) and loratadine were procured from Caplin Point Laboratories Ltd. Chennai, as a gift sample (India). Carbopol 934, sodiumcarboxy methylcellulose (Na CMC), chitosan, propylene glycol (PG), polyethylene glycol (PEG), mannitol, xanthan gum, methanol, benzalkonium chloride, and Hydroxypropyl methylcellulose (HPMC K 100) were procured from SD Fine Chemicals, Bangalore (India). EDTA, sodium taurocholate, Pluronic F 127, and oleic acid were obtained from Loba Chemicals, Mumbai, India. All chemicals and reagents used were of analytical grade.

### 2.2. Methods

#### 2.2.1. Formulation of In Situ Nasal Gels without Permeation Enhancers

A 2^4^ factorial design was used for the optimization of process parameters. Table 1 shows the composition of different formulations of nasal gels (A–F). The drugs were dissolved in methanol and added to 10 mL of distilled water (Milli-Q) with constant stirring in separate small beakers. To the first beaker, the above drug loratadine solution (PEG, mannitol, and benzalkonium chloride) was added and, to the second beaker, a CPM solution (mannitol, PG, and benzalkonium chloride) was added. The polymeric solutions of HPMC K100 and xanthan gum were made separately in distilled water and thoroughly mixed with the above loratadine mixture. The polymeric solutions of chitosan, carbopol 934, and Na CMC were prepared separately using distilled water and then mixed to form the above CPM mixture in the second beaker. The resultant drug mixtures were magnetically stirred for 15 min, and then a phosphate buffer solution was added. Distilled water was used to adjust the final volume to the desired amount [23].

#### 2.2.2. Evaluation of In Situ Nasal Gels

The prepared in situ nasal gels (A–F) were evaluated for gelling temperature, gelling time, viscosity, rheological properties, pH, drug content, gel strength, spreadability, mucoadhesive strength, in vitro drug permeation, and in vitro drug dissolution. The procedures were reported in our earlier publications [24,25].

##### Gelling Temperature

An adequate quantity of the prepared solutions was placed in a test tube, which was then placed on a thermostat set at four degrees Celsius. The temperatures were then gradually increased at a rate of 1 °C every two minutes.

##### Gelling Time

For gelling time, the transition temperature of the prepared formulations (A–F) of in situ gels was determined by transferring a two-milliliter formulation to a test tube (TT), with a diameter of one centimeter. After sealing the TT by parafilm, it was placed in a water bath at 37 °C. After each temperature setting, equilibration was permitted for 10 min. Finally, the TT was kept in the horizontal position to see the nature of the sample and also to examine the conversion of sol to gel.

##### Viscosity of the Solution

It was evaluated using a Brookfield viscometer DV-II+ paired with an S-94 spindle. The prepared gel formulations (A–F) were transferred to the beaker, and the spindle was lowered perpendicularly into the gel at 100 rpm, at 37 ± 0.5 °C. During the cooling of the system, the parameter was determined. All the measurements were performed in triplicate.

##### Flow Properties of the In Situ Gels

The flow properties were evaluated using a Brookfield LVDV-E Viscometer. Initially, the temperature was maintained above 40 °C. By varying the spindle speed from 0.3 to 100 rpm, the viscosity (h), the rate of shear (g), and shearing stress (t) were observed (measurements were performed in triplicate).

##### Drug Content Determination

To analyze the content of the formulation, 1 mL of the prepared formulation was dispersed in 10 mL methanol for 2–3 min while shaking occasionally. The mixture was then filtered through 0.45 μm filter paper and diluted. The concentration of loratadine in the developed formulation was measured with a UV spectrophotometer at a wave length of 280 nm.

##### Gelation Strength

Gelation strength was determined by keeping each of the formulations (A–F) on a thermostat at 37 °C. A 50 g sample was introduced in a measuring cylinder of 100 mL capacity and the gel strength was determined by measuring the time for a weight of 35 g to sink 5 cm through the sample.

##### pH Determination

All prepared formulations were tested for pH. A total of 1 milliliter of the developed gels was placed in a 10-milliliter standard volumetric flask, diluted with distilled water up to 10 mL. The pH of the resulting formulation was measured by a digital pH meter (ShambhaviImpex, Navi Mumbai, India).

##### Spreadability

A 10 × 4 cm rectangular glass slide was used to test the spreadability. The nasal mucosa of the sheep from the serosal side was threaded onto the surface of the slide. The slide was heated in the oven at 37 °C, and one drop of gel was inserted on the mucosa at a 120° angle. The spreadability was calculated by measuring the distance covered by a drop of liquid gel prior to gelation.

##### Ex Vivo Mucoadhesive Strength

Fresh sheep nasal mucosa was used to measure the ex vivo mucoadhesive strength. A Franz diffusion cell was used to determine in vitro drug release. The mechanism by which the drug can be released from the formulation was determined by plotting the release data’s best fit in Korsmeyer–Peppas and Higuchi plots. With the help of the linear regression method and employing Microsoft Excel 2003 software, k- and n-release rate constants were determined. The coefficient of determination (R^2^) was utilized to determine the accuracy.

### 2.3. Drug–Polymer Interaction Studies

With respect to preformulation studies, FTIR, DSC, and XRD were used for the interaction analysis of drug polymer for the optimized formulation of the in situ nasal gel [26,27].

### 2.4. Effect of Various Permeation Enhancers in the Permeation of Loratadine and Chlorpheniramine Maleate from the In Situ Nasal Gels

In vitro studies on nasal permeation were performed on the in situ nasal gels with and without permeation enhancers. The in vitro nasal permeation of loratadine and chlorpheniramine maleate was studied through the sheep nasal mucosa by the vertical diffusion cell method (1 cm^2^) thermostated at 37 °C in a water bath (Variomag, Seelbach, Germany). Freshly obtained nasal membrane was placed between the receptor and donor compartments, in which the smoothened portion faced the donor side. The sample was kept on the membrane and both sides cinched together. The donor compartment was slightly wetted with one mL of buffer (pH 6.4), and the other was filled with an isotonic buffer (pH 7.4). This was conducted thusly because the drug can infuse the nasal mucosa and attain systemic circulation (pH 7.4). However, to contain the nasal pH, the gels were wetted prior to the permeation study, with a buffer pH of 6.4. The receptor portions were stirred at 600 rpm. A total of 1 mL of the solution was taken at different time intervals (0, 0.5, 1, 2, 4, 6, 8, 10, and 12 h), which was substituted with a blank. This was then filtered, and after suitable dilution, the analysis was carried out with an ultraviolet spectrophotometer.

The in situ nasal gels with their formulation codes are shown in Table 2. The permeation enhancers’ concentration in each in situ nasal gels were EDTA (0.2% *w*/*v*), sodium taurocholate (0.5% *w*/*v*), oleic acid (5% *w*/*v*), and Pluronic F 127 (10% *w*/*v*).

## 3. Results and Discussion

### 3.1. Evaluation of In Situ Gels

#### 3.1.1. Gelling Temperature and Time, Drug Content, Gel Strength, and Viscosity of Solution

The in situ gels of loratadine and chlorpheniramine maleate were evaluated for gelling temperature and time, gel strength, drug content, and viscosity of the solution. The physicochemical properties of the prepared in situ nasal gels of loratadine and chlorpheneramine maleate are presented in Table 3.

The gelling temperature of the formulated loratadine in situ nasal gel varied from 33.4 ± 0.83 °C to 34.8 ± 0.82 °C and that of chlorpheniramine maleate showed a range of 31.1 ± 0.46 °C to 31.2 ± 0.12 °C. The structure of the resulting sample depends on the amount of polymer used. The gelling temperature is the temperature at which the liquid transitions into gel. The gel’s gelling temperature varies in the range of 30–36 °C. The gelling point relates to the temperature: the meniscus of the formulation stops moving when the test tubes are tilted at 90°, with a gradual increase in temperature. The gels can be formed at RT, when the gelling temperature is less than the referred range, though the gelation does not occur in the nasal mucosal area, resulting in a quick nasal clearance [28]. However, all of the developed formulations displayed long-term integrity for sustained drug release and rapid gelation when touched with the solution. The formulation’s gelling time (s) varied from 4.0 ± 0.21 s to 11.3 ± 0.22 s. The gelling time is the time it takes for a sol to form a gel. The drug content of the prepared loratadine formulation was between 97.78 ± 0.54% and 99.76 ± 0.12% and that of the chlorpheniramine maleate formulation was between 99.12 ± 0.32% and 99.42 ± 0.43%. The other key characteristics of the mucoadhesive in situ nasal gel systems are viscosity and gel capacity. The viscosity of the developed loratadine formulation varied between 180.24 ± 0.54 cP and 240.76 ± 1.67 cP and that of the chlorpheniramine maleate formulation ranged from 191.21 ± 0.11 cP to 280.23 ± 1.18 cP. The viscosity is directly correlated with the compositions of the polymeric content [29]. When the phosphate buffer solution was used to mix with in situ gels, a viscous gel was instantaneously designed. When the formulations were prepared, the viscosity was increased.

The gel strength of the loratadine formulation ranged between 56.99 ± 0.42 and 62.12 ± 0.52 and that of the chlorpheniramine maleate formulation ranged between 73.53 ± 0.32 and 83.12 ± 0.22. A mucoadhesive system designed for nasal drug delivery should maintain its integrity and clarity without eroding/dissolving for an extended period of time to ensure controlled drug release on tissues. However, here, all formulations gelled immediately upon contact with the simulated nasal fluids. The gelling time of the formulations decreased in highly viscous formulations. Thus, the gel strength of the formulations was higher in highly viscous formulations [30].

#### 3.1.2. Ph, Spreadability, and Mucoadhesive Strength

The physicochemical properties, such as pH, mucoadhesive strength, and spreadability, are shown in Table 4. The pH of the loratadine formulation was between 5.7 ± 0.004 and 6.0 ± 0.003 and that of the chlorpheniramine maleate formulation varied from 5.7 ± 0.007 to 5.8 ± 0.008. These results reveal that the formulations’ pH was within acceptable ranges and did not cause mucosal irritation. The spreadability of the loratadine formulation was between 7.0 ± 0.82 and 7.2 ± 0.76 cm and that of the chlorpheniramine maleate formulation ranged between 5.8 ± 0.86 and 6.9 ± 0.12. Formulations prepared with high levels of polymers showed excellent spreadability. The results indicate that the spreadability of the prepared formulations was lower for highly viscous formulations. The mucoadhesive property increased when the concentration of polymers used also increased [31]. The mucoadhesive strength of the loratadine formulations ranged from 6498.98 ± 0.56 to 6561.56 ± 0.98 dyne/cm^2^, and that of the chlorpheniramine maleate formulations ranged between 7865.38 ± 0.48 and 8378.54 ± 0.36 dyne/cm^2^. Mucoadhesive drug delivery methods aid in the rapid absorption of active drugs in the circulatory system, preventing first-pass metabolism, and increasing the length of residency at the site of application [32].

#### 3.1.3. In Vitro Drug Release

The in vitro drug release of all developed formulations (A–F) is shown in Figure 1. The loratadine formulations A, B, and C showed drug release after 10 h and that of chlorpheniramine maleate formulations D, E, and F was up to 12 h. During the study period, the formulations made with high-viscosity polymers prolonged drug release. The concentration and viscosity of the polymer used corresponded with the release of different samples.

The formulation A was composed of loratadine (7.5 mL), xanthan gum (0.1 mL), HPMC K 100 (0.8 mL), PEG (1.25 mL), mannitol (2.5 mL), benzalkonium chloride (0.01), and methanol (5 mL). The formulation B was composed of loratadine (7.5 mL), xanthan gum (0.15 mL), HPMC K 100 (0.8 mL), PEG (1.25 mL), mannitol (2.5 mL), benzalkonium chloride (0.01), and methanol (5 mL). The formulation C was composed of loratadine (7.5 mL), xanthan gum (0.25 mL), HPMC K 100 (0.2 mL), PEG (1.25 mL), mannitol (2.5 mL), benzalkonium chloride (0.01), and methanol (5 mL). The loratadine formulations A (99.87%), B (99.76%) and C (99.98%), showed drug release after 10 h. The rate of release was associated with the concentration of HPMC, at fixed drug doses. Drug release was lower when the HPMC concentration was high. Every sample was a solid match to the Higuchi model. As indicated by this method, the release of drug is diffusion-dependent, which depends on Fick’s laws, depicting the transport across conc gradient. The release from the polymer matrices was affected by erosion.

The formulation Dwas composed of chlorpheniraminemaleate (7.5 mL), Carbopol 934 (0.1 mL), Na CMC (0.5 mL), PG (1 mL), chitosan (1 mL), mannitol (2.5 mL), and benzalkoniumchloride (0.01). The formulation Ewas composed of chlorpheniraminemaleate (7.5 mL), Carbopol 934 (0.2 mL), Na CMC (0.5 mL), PG (1 mL), chitosan (1 mL), mannitol (2.5 mL), and benzalkoniumchloride (0.01 mL). The formulation F was composed of chlorpheniraminemaleate (7.5 mL), Carbopol 934 (0.3 mL), Na CMC (0.5 mL), PG (1 mL), chitosan (1 mL), mannitol (2.5 mL), and benzalkoniumchloride (0.01 mL). The chlorpheniramine maleate formulations D (99.98%), E (99.99%), and F (99.98%) showed drug release up to 12 h. All samples showed a best fit with the Higuchi method and followed a diffusion mechanism. As per the method, drug released from prepared ones could be controlled by means of diffusion through micropores. Diffusion follows Fick’s laws, and it explains the passage of molecules from a lower to a higher concentration.

Drug release from the developed in situ nasal formulations (A–F) was found using the best fit model of data release from Higuchi and Korsmeyer–Peppas plots. According to Higuchi’s equation, Qt = K_H_ √t, where Qt is the amount of drug released at time t and K_H_ is Higuchi’s rate constant, whereas Korsmeyer–Peppas’s equation shows Q_t_/Q_α_ = Kt^n^, where Q_t_/Q_α_ is the portion of the drug released at time t, K is the release rate constant, and n is the exponential that characterizes the drug release mechanism. The profile and kinetics of drug release are important because they correlate the in vitro response by comparing the result of the dissolution profile pattern. Different mathematical models may be applied to describe the kinetics of the drug release process from the formulation; the most suited is the one that best fits the experimental results. These models best describe drug release from pharmaceutical systems resulting from a simple phenomenon or when this phenomenon, by being the rate-limiting step, conditions all the other processes occurring in the system.

The release of the drug from the in situ nasal formulations was found using the best fit model of data release from Higuchi and Korsmeyer–Peppas plots. All samples showed a best fit with the Higuchi method and followed a non-Fickian diffusion mechanism. The n value was between 0.775 and 1.910 (an n value less than 1 indicates non-Fickian anomalous type 1 release) [33]. The values of r^2^, n, and k values are depicted on Table 5. As per the method, the drug released from the prepared ones could be controlled by means of diffusion through micropores. Diffusion follows Fick’s laws, and it explains the passage of molecules from a lower to a high concentration. Matrix erosion may also affect the drug release from the polymer matrix. Release can be owed to a combination erosion and diffusions. The mechanism of release from controlled devices seems complex. Several drug release mechanisms are related to diffusion or otherwise erosion, but some follows diffusion and erosion [34]. The diffusion of drug takes place through a solvent-filled pathway of gels, especially when swelling is predominant.

#### 3.1.4. Drug–Polymer Interaction Studies of the Selected Formulations

The principal peak at or around the required drug wave numbers were expressed by the FTIR spectra of the formulation of loratadine (A) and formulation of chlorpheniramine (D). Hence, it was found that there was not a chemical interaction between polymers and drugs, and its purity and integrity was upheld in the formulation, as shown in Figure 2a,b. The DSC thermogram of the developed formulations A and D expressed no interaction with the polymers used, as shown in Figure 3a,b. The XRD of the selected formulations A and D showed that the observed peaks in the pure drug and the formulation did not interact, as shown in the Figure 4a,b. All interaction studies with FTIR, DSC, and XRD proved there was no interaction of selected excipients with the pure drug.

#### 3.1.5. Effect of Various Permeation Enhancers on the Infusion of Loratadine and Chlorpheniramine Maleate across Sheep Nasal Mucosa

The permeation data of loratadine and chlorpheniramine maleate infused from the selected in situ nasal gels through sheep nasal mucosa (with and without permeation enhancers) are shown in Figure 5, Figure 6 and Figure 7.

The formulations A and B did not have permeation enhancers. The formulations AP1 and BP1 were composed of EDTA (0.2% *w*/*v*); AP2 and BP2 were composed of sodium taurocholate (0.5% *w*/*v*); AP3 and BP3 were composed of Pluronic F127 (10% *w*/*v*); and AP4 and BP4 were composed of oleic acid (5% *w*/*v*) as permeation enhancers.

The formulations C and D did not have permeation enhancers. The formulations CP1 and DP1 were composed of EDTA (0.2% *w*/*v*); CP2 and DP2 were composed of sodium taurocholate (0.5% *w*/*v*); CP3 and CP3 were composed of Pluronic F127 (10% *w*/*v*); and CP4 and DP4 were composed of oleic acid (5% *w*/*v*) as permeation enhancers.

The formulations E and F did not have permeation enhancers. The formulations EP1 and FP1were composed of EDTA (0.2% *w*/*v*); EP2 and FP2 were composed of sodium taurocholate (0.5% *w*/*v*); EP3 and FP3 were composed of Pluronic F127 (10% *w*/*v*); and EP4 and FP4 were composed of oleic acid (5% *w*/*v*) as permeation enhancers.

Comparing the permeation data of the formulations (by using and not using PE) indicates that these formulations produced an improved penetration across the nasal membrane with the support of permeation enhancers. The permeation data for loratadine and chlorpheniramine maleate through sheep nasal mucosa are shown in Table 6.

The best course of action is to introduce a permeation enhancer in order to increase drug absorption into the systemic circulation. Due to mucoadhesion and the temporary opening of the cell membrane’s tight junctions, polar medications can pass through a para-cellular channel. The effectiveness and safety of nasal absorption enhancers depend on a number of variables, such as their effect on nasal epithelial membrane barriers, the level of enzymatic activity in the nasal cavity and mucociliary clearance. The use of bioadhesive polymers that can adhere to the nasal mucosa for a sufficient period of time and inhibit rapid nasal clearance has been found advantageous [35,36].

Among these permeation enhancers, sodium taurocholate, Pluronic F127, and oleic acid produced a noticeable increase in the flux of loratadine in situ nasal gels compared with in situ nasal gels without PE, while adding only EDTA slightly enhanced the flux, but, in most cases, the enhancement was insignificant. However, in the case of chlorpheniramine maleate in situ nasal gels, the permeation enhancer oleic acid only showed a noticeable increase in flux.

Sodium taurocholate and oleic acid seem to be better and more efficient enhancers, enhancing the flux > 5-fold compared with in situ nasal gels without the permeation enhancer in loratadine in situ nasal gels. Pluronic F127 also demonstrated improved permeation, increasing the effect of loratadine in situ by more than twofold. In chlorpheniramine maleate in situ nasal gels, EDTA, sodium taurocholate, and Pluronic F127 were equally effective in enhancing chlorpheniramine maleate permeation. Oleic acid works better as a permeation enhancer in chlorpheniramine maleate in situ nasal gels and showed a maximum permeation enhancement of more than 2-fold.

The paracellular transport of drugs is enhanced by bile salts, such as sodium taurocholate. This occurs due to the opening of tight mucosal junctions due to bile salt-induced calcium complexation. However, this change is temporary and reversible. The method by which permeation enhancers (fatty acids) improve drug penetration through nasal membranes is not well known. However, the processes are expected to be the same as those used to improve skin permeability. Oleic acid and Pluronic F127 may interact with lipids and destabilise their structures, resulting in enhanced fluidity and flux. Higher enhancement factors were obtained by using oleic acids that contain one double bond. The enhancement ratio decreased by the presence of additional double bonds.

## 4. Conclusions

To bypass first-pass metabolism and improve the drug’s consequently limited bioavailability, novel mucoadhesive in situ gels containing loratadine and chlorpheniramine maleate were produced. The in vitro investigations revealed that the in situ gels can be used as a possible drug delivery vehicle for loratadine, with improved stability and release profiles. The physicochemical properties of all of the formulations studied were comparable. Permeation is further customised by employing various permeation enhancers. When compared to in situ gels without permeation enhancers, all permeation enhancers generated a significant increase in loratadine and chlorpheniramine flux. Sodium taurocholate, Pluronic f127, and oleic acid generated a significant increase in the flux of the loratadine in situ nasal gel as compared to in situ nasal gels without permeation enhancers, although EDTA just marginally enhanced the flux, but, in most cases, the enhancement was insignificant. However, in the scenario of chlorpheniramine maleate in situ nasal gels, the permeation enhancer oleic acid only demonstrated a substantial increase in flux. Future investigations utilising formulations with these potential permeation enhancers are necessary to validate these results in vivo.

## Figures and Tables

**Figure 1 polymers-15-01206-f001:**
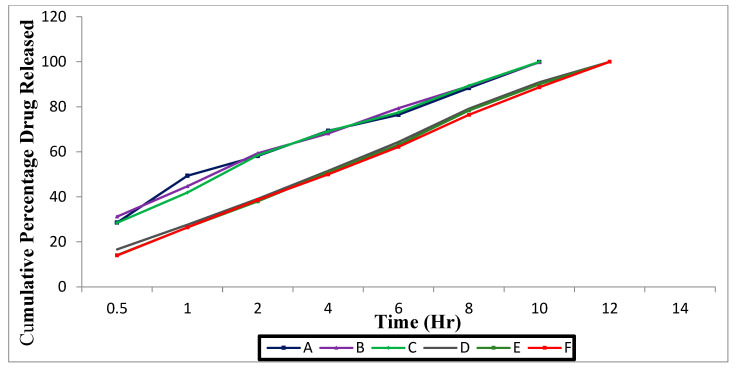
In vitro drug release of the in situ nasal gel formulations: A, B, C, D, E, and F.

**Figure 2 polymers-15-01206-f002:**
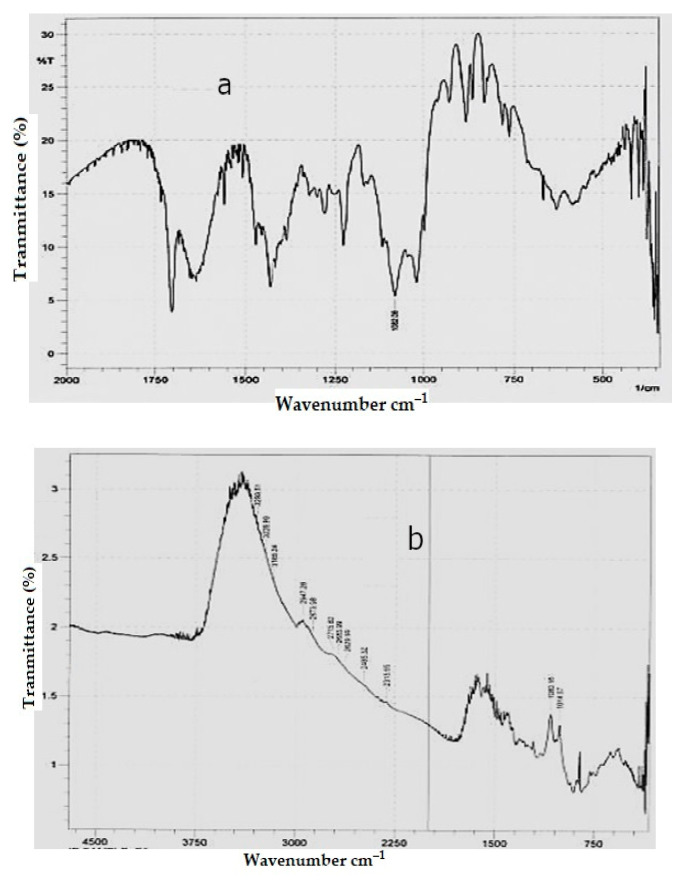
Spectra of the FTIR of the selected formulation of loratadine (A) and chlorpheniramine (D) are depicted in (**a**,**b**), respectively.

**Figure 3 polymers-15-01206-f003:**
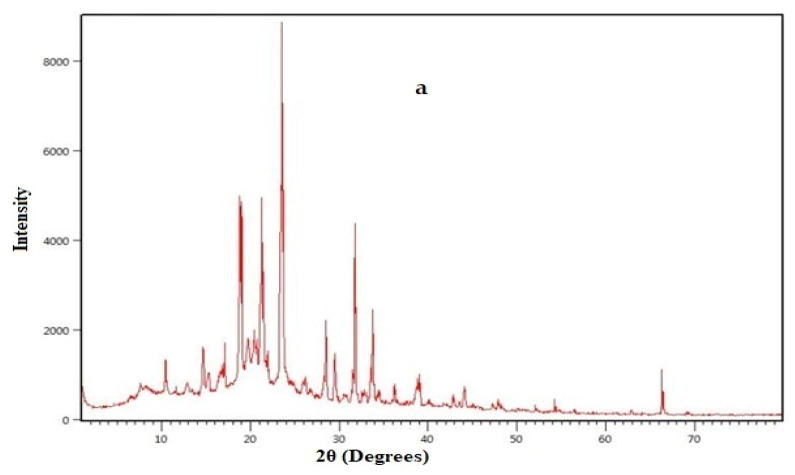
XRD spectra of the selected formulations of loratadine (A) and chlorpheniramine (D) are depicted in (**a**,**b**), respectively.

**Figure 4 polymers-15-01206-f004:**
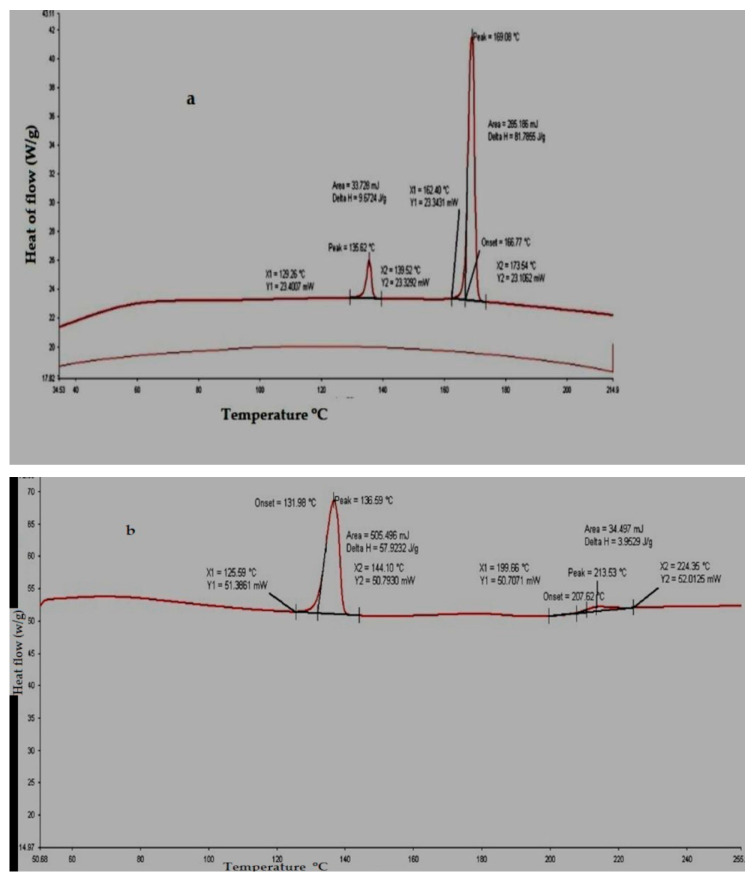
DSC thermograms of the selected formulations of loratadine (A) and chlorpheniramine (D) are depicted in (**a**,**b**), respectively.

**Figure 5 polymers-15-01206-f005:**
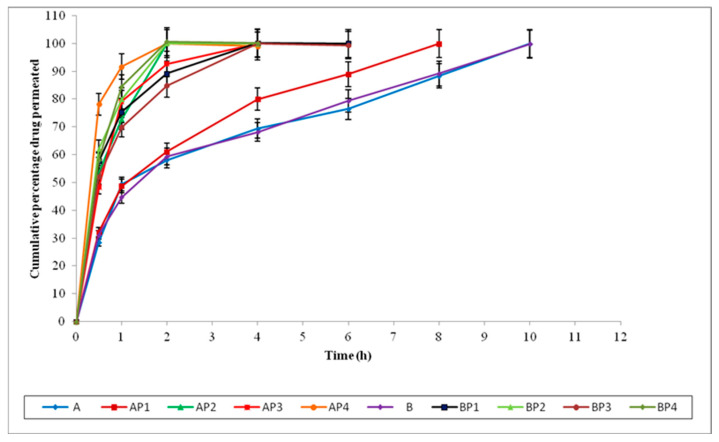
Percentage of drug infused from in situ nasal gel formulations (A, AP1, AP2, AP3, AP4, B, BP1, BP2, BP3, and BP4) with and without PE across sheep nasal mucosa.

**Figure 6 polymers-15-01206-f006:**
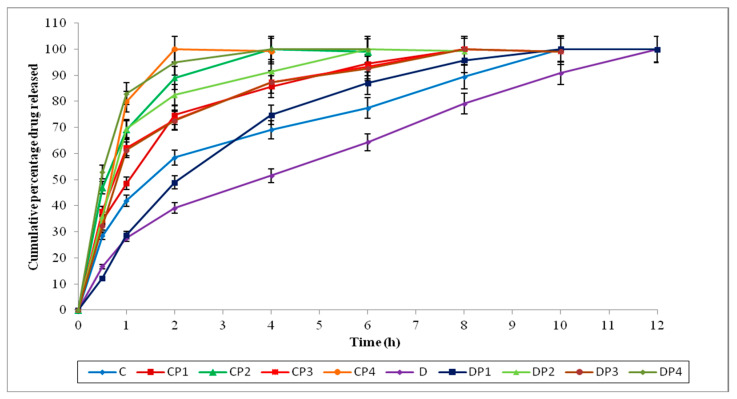
Percentage of drug infused from in situ nasal gel formulations (C, CP1, CP2, CP3, CP4, D, DP1, DP2, DP3, and DP4) with and without PE across sheep nasal mucosa.

**Figure 7 polymers-15-01206-f007:**
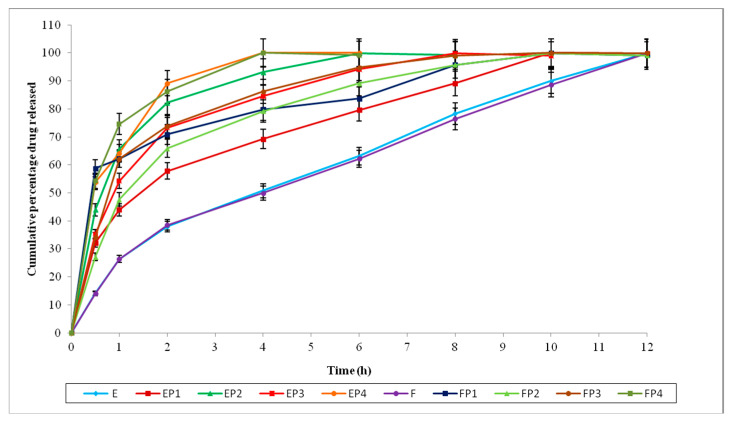
Percentage of drug permeated from the in situ nasal gel formulations (E, EP1, EP2, EP3, EP4, F, FP1, FP2, FP3, and FP4) with and without PE across sheep nasal mucosa.

**Table 1 polymers-15-01206-t001:** Composition of the loratadine and chlorpheniramine maleate in situ nasal gels with chlorpheniramine maleate, xanthan gum, HPMC K 100, Carbopol 934, Na CMC, and chitosan as polymers and without permeation enhancers.

Composition	Formulation Code					
	A	B	C	D	E	F
	F4	F8	F13	G4	G8	G12
Loratadine (%*w*/*v*)	7.5	7.5	7.5	-	-	-
Chlorpheniramine maleate (%*w*/*v*)	-	-	-	7.5	7.5	7.5
Xanthan gum (%*w*/*v*)	0.1	0.15	0.25	-	-	-
HPMC K 100 (%*w*/*v*)	0.8	0.8	0.2			
Carbopol 934 (%*w*/*v*)	-	-	-	0.1	0.2	0.3
Na CMC (%*w*/*v*)	-	-	-	0.5	0.5	0.5
Chitosan (%*w*/*v*)	-	-	-	1.0	1.0	1.0
PEG (%)	1.25	1.25	1.25	-	-	-
PG (%)	-	-	-	1	1	1
Mannitol (%*w*/*v*)	2.5	2.5	2.5	2.5	2.5	2.5
Benzalkonium chloride (%*w*/*v*)	0.01	0.01	0.01	0.01	0.01	0.01
Methanol (mL)	5	5	5	-	-	-
Phosphate buffer solution (pH 6.4) (mL)	2.5	2.5	2.5	-	-	-
Distilled water (mL; QS)	25	25	25	25	25	25

**Table 2 polymers-15-01206-t002:** In situ nasal gels with permeation enhancers.

Formulation with PE	Permeation Enhances	Formulation Code
A	EDTA (0.2% *w*/*v*)	AP1
	Sodium taurocholate (0.5% *w*/*v*)	AP2
	Pluronic F 127 (10% *w*/*v*)	AP3
	Oleic acid (5% *w*/*v*)	AP4
B	EDTA (0.2% *w*/*v*)	BP1
	Sodium taurocholate (0.5% *w*/*v*)	BP2
	Pluronic F 127 (10% *w*/*v*)	BP3
	Oleic acid (5%)	BP4
C	EDTA (0.2% *w*/*v*)	CP1
	Sodium taurocholate (0.5% *w*/*v*)	CP2
	Pluronic F 127 (10% *w*/*v*)	CP3
	Oleic acid (5% *w*/*v*)	CP4
D	EDTA (0.2% *w*/*v*)	DP1
	Sodium taurocholate (0.5% *w*/*v*)	DP2
	Pluronic F 127 (10% *w*/*v*)	DP3
	Oleic acid (5% *w*/*v*)	DP4
E	EDTA (0.2% *w*/*v*)	EP1
	Sodium taurocholate (0.5% *w*/*v*)	EP2
	Pluronic F 127 (10% *w*/*v*)	EP3
	Oleic acid (5% *w*/*v*)	EP4
F	EDTA (0.2% *w*/*v*)	FP1
	Sodium taurocholate (0.5% *w*/*v*)	FP2
	Pluronic F 127 (10% *w*/*v*)	FP4
	Oleic acid (5% *w*/*v*)	FP3

**Table 3 polymers-15-01206-t003:** The physicochemical properties of the prepared in situ nasal gels of loratadine and chlorpheneramine maleate.

Formulation Code	Gelling Temperature (°C) *	Gelling Time (s) *	Viscosity of Solution (cP) *	Drug Content (%) *	Gel Strength (s) *
A	34.2 ± 0.82	8.0 ± 0.53	180.24 ± 0.54	99.76 ± 0.12	60.32 ± 0.43
B	33.6 ± 0.47	7.8 ± 0.76	220.67 ± 1.21	98.56 ± 0.32	62.12 ± 0.52
C	33.4 ± 0.83	4.0 ± 0.21	240.76 ± 1.67	97.78 ± 0.54	56.99 ± 0.42
D	31.1 ± 0.46	6.8 ± 0.28	191.21 ± 0.11	99.12 ± 0.32	73.53 ± 0.32
E	31.2 ± 0.12	5.8 ± 0.21	242.52 ± 1.18	99.42 ± 0.43	78.45 ± 0.46
F	31.1 ± 0.56	4.6 ± 0.72	280.23 ± 1.18	99.32 ± 0.12	83.12 ± 0.22

* Mean ± SD, *n* = 3.

**Table 4 polymers-15-01206-t004:** The physicochemical properties, such as pH, spreadability, and mucoadhesive strength.

Formulation Code	pH	Spreadability (cm)	Mucoadhesive Strength (dyne/cm^2^)
A	6.0 ± 0.002	7.1 ± 0.16	6542.56 ± 0.56
B	6.0 ± 0.003	7.2 ± 0.76	6498.98 ± 0.56
C	5.7 ± 0.004	7.0 ± 0.82	6561.56 ± 0.98
D	5.7 ± 0.007	6.9 ± 0.12	7865.38 ± 0.48
E	5.7 ± 0.007	6.5 ± 0.78	8120.12 ± 0.11
F	5.8 ± 0.008	5.8 ± 0.86	8378.54 ± 0.36

**Table 5 polymers-15-01206-t005:** Coefficient values of the different formulations of the nasal in situ gels of loratadine and chlorpheniramine maleate.

Formulations	Higuchi			Korsmeyer–Peppas			Mechanism
	r^2^	y	K	r^2^	y	n	
A	0.959	28.97x + 9.639	28.97	0.323	0.778x + 1.253	0.778	Higuchi
B	0.971	29.31x + 9.268	29.31	0.321	0.775x + 1.256	0.775	Higuchi
C	0.977	29.81x + 7.540	29.81	0.342	0.800x + 1.236	0.800	Higuchi
D	0.9948	28.801x − 2.3504	28.801	0.5038	0.8799x + 1.0882	0.8799	Higuchi
E	0.9929	28.977x − 3.627	28.977	0.5335	0.9102x + 1.0594	0.9102	Higuchi
F	0.9914	28.603x − 3.4252	28.603	0.5315	0.906x + 1.0582	0.906	Higuchi

**Table 6 polymers-15-01206-t006:** The data of the permeation of loratadine and chlorpheniramine maleate through sheep nasal mucosa.

Formulation Code	Steady State Flux mg·cm^2^·min^−1^	Permeability Coefficient cm^2^·min^−1^	Enhancement Ratio
A (without permeation enhancers)	7.926	10.568	1
AP1 (0.2% *w*/*v*)	10.473	13.964	1.3213
AP2 (0.5% *w*/*v*)	46.418	61.890	5.8563
AP3 (10% *w*/*v*)	20.856	27.808	2.6313
AP4 (5% *w*/*v*)	43.272	57.696	5.4595
B (without permeation enhancers)	8.0384	10.717	1
BP1(0.2% *w*/*v*)	19.962	26.616	2.4835
BP2 (0.5% *w*/*v*)	45.339	60.452	5.6407
BP3 (10% *w*/*v*)	20.528	27.370	2.5538
BP4 (5% *w*/*v*)	46.597	62.129	5.7972
C (without permeation enhancers)	8.2185	10.958	1
CP1 (0.2% *w*/*v*)	10.501	14.001	1.2776
CP2 (0.5% *w*/*v*)	21.296	28.394	2.5911
CP3 (10% *w*/*v*)	9.8245	13.099	1.1953
CP4 (5% *w*/*v*)	50.026	66.701	6.0896
D (without permeation enhancers)	7.6126	10.150	1
DP1 (0.2% *w*/*v*)	9.8555	13.140	1.2945
DP2 (0.5% *w*/*v*)	13.559	18.078	1.7810
DP3 (10% *w*/*v*)	10.095	13.46	1.3261
DP4 (5% *w*/*v*)	20.318	27.090	2.6689
E (without permeation enhancers)	7.6828	10.243	1
EP1 (0.2% *w*/*v*)	8.0793	10.772	1.0516
EP2 (0.5% *w*/*v*)	13.338	17.784	1.7362
EP3 (10% *w*/*v*)	10.226	13.634	1.3310
EP4 (5% *w*/*v*)	20.891	27.854	2.7193
F (without permeation enhancers)	7.5893	10.119	1
FP1 (0.2% *w*/*v*)	6.6922	8.922	0.8817
FP2 (0.5% *w*/*v*)	8.4635	11.284	1.1151
FP3 (10% *w*/*v*)	8.1233	10.831	1.0703
FP4 (5% *w*/*v*)	20.153	26.870	2.6554

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
