# Peer review of "Combination of Cellulose Derivatives and Chitosan-Based Polymers to Investigate the Effect of Permeation Enhancers Added to In Situ Nasal Gels for the Controlled Release of Loratadine and Chlorpheniramine"

_polymers, 2023, doi:10.3390/polym15051206_

Round 1

Reviewer 1 Report

In this manuscript, Vasantha and coworkers study the effect of the permeation enhancers in the in situ nasal gels of loratadine and chlorpheniramine. EDTA, sodium taurocholate, Oleic acid, and Pluronic F 127 on the nasal absorption of loratadine and chlorpheniramine from in situ nasal gels containing different polymeric combinations including hydroxypropyl methylcellulose, carbopol 934, sodium carboxy methylcellulose, and chitosan are studied. Such studies show good potential to develop nano-carriers for drug delivery in vivo. Therefore, I would recommend this manuscript to be published on Polymers after minor revision.

1. I think the author should clearly discuss or demonstrate the relationship between their study and the effect of drug delivery to improve the application potential of this manuscript.

2. Also, I need to emphasize that improving the quality of Figures (Figure 2, 3, 4) and reorganizing them is very important not only on better exhibition, but also on better academic logic. I think only the exhibition of original data without further organization is not suitable for publication.

3. The authors may consider citing the latest progress on drug delivery, such as ACS Appl. Mater. Interfaces 2018, 10, 24987.

Author Response

Authors are highly thankful to esteemed reviewers for their excellent comments for further improvement of the article.

Responses to questions raised by esteemed reviewers are provided below in a point-by-point format. All revisions are highlighted in the revised manuscript.

Reviewer 2 Report

This article aims to increase the permeability of different substances to the nasal delivery. The article is well design and the experiments well explained. The novelty is doubtful due to the fact there is no comparison with other similar studies. I would suggest to add a table or a paragraph where you discuss your results in comparison with other ones. Therefore, the introduction needs to be highly improved also on the nasal delivery. Please add the following references and much more.

Kaur, P., Garg, T., Rath, G. and Goyal, A.K., 2016. In situ nasal gel drug delivery: A novel approach for brain targeting through the mucosal membrane. Artificial cells, nanomedicine, and biotechnology44(4), pp.1167-1176.

Türker, S., Onur, E. and Ózer, Y., 2004. Nasal route and drug delivery systems. Pharmacy world and Science26(3), pp.137-142.

Varshosaz, J., Sadrai, H. and Heidari, A., 2006. Nasal delivery of insulin using bioadhesive chitosan gels. Drug delivery13(1), pp.31-38.

Baldelli, A., Boraey, M.A., Oguzlu, H., Cidem, A., Rodriguez, A.P., Ong, H.X., Jiang, F., Bacca, M., Thamboo, A., Traini, D. and Pratap-Singh, A., 2022. Engineered nasal dry powder for the encapsulation of bioactive compounds. Drug Discovery Today.

Author Response

(The authors gave the same response as above.)

Reviewer 3 Report

The Manuscript entitled "Effect of Permeation Enhancers in the In Situ Nasal Gels of Loratadine and Chlorpheniramine" can not be aacapted by Polymers Journal. The fact that the authors used multiple polymers to form the nasal gel does not indicate research in the field of polymers. The article would be suitable for another MDPI Journal, for example, Pharmaceutics, however, there are many questions about the composition of the nasal gel, such as presence of methanol in it, which is used to dissolve the drugs. 

Bearing the aforementioned in mind, I recommend to reject the paper. 

Author Response

Dear reviewer, we have extensive revised and updated, which are necessary for the manuscript. we have thoroughly checked the manuscript for any incurred grammar, spelling/syntax errors and rectifications have been made at respective places in the revised manuscript. 

We authors kindly request to the reviewer to consider our revised/updated manuscript for publications.
